# Effects of the Task Complexity on the Single Movement Response Time of Upper and Lower Limbs in Police Officers

**DOI:** 10.3390/ijerph19148695

**Published:** 2022-07-17

**Authors:** Dunja Janković, Aleksandar Čvorović, Milivoj Dopsaj, Iva Prćić, Filip Kukić

**Affiliations:** 1Abu Dhabi Police, Police Sports Education Center, Abu Dhabi 253, United Arab Emirates; dunja.prcic12@gmail.com (D.J.); cvorovic77@yahoo.com (A.Č.); 2Faculty of Sport and Physical Education, University of Belgrade, 11030 Belgrade, Serbia; milivoj.dopsaj@fsfv.bg.ac.rs (M.D.); iva.prcic@yahoo.com (I.P.); 3Institute of Sport, Tourism and Service, South Ural State University, 454080 Chelyabinsk, Russia; 4Institute of Medical Research, Belgrade University, 11129 Belgrade, Serbia

**Keywords:** reaction speed, occupational performance, law enforcement officers

## Abstract

Police officers occasionally encounter belligerents resisting or even physically assaulting them without or with objects. The self-defense or legal utilization of use of force to disable the offender from harming an officer or others may depend on a single movement speed of hands and legs. This study investigated the effects of task complexity on a single movement response time of the upper and lower limbs in police officers. The sample consisted of 32 male police officers aged between 23 and 50 years. They performed a single movement as fast as possible with their upper and lower limb in three incrementally more complex tasks. In the first task, participants acted on a light signal and with their dominant limb they had to turn off the signal as fast as possible. In the second task, on the light signal, participants could turn off the light with free choice of the upper limb in a hand task or lower limb in a leg task. In the third task, participants had to turn the light off with the right limb if the light turned red and with the left limb if the light turned blue. The BlazePod device was used to assess the movement response time. The results show that there was a significant effect of task complexity on the single movement response time of the hand (F = 24.5, *p* < 0.001) and leg (F = 46.2, *p* < 0.001). The training of police officers should utilize specific and situational tasks to improve movement response time by improving the redundancy in decision-making processes during work-specific tasks of different complexity.

## 1. Introduction

Speed is an essential component of physical fitness that helps an individual to perform well in a large number of sports [1]. In martial arts and police occupation, successful task performance largely depends on the fast and dynamic movement of an individual (e.g., kick, block, catch-and-grip, discharge weapon) [2,3]. The ability to reach maximum reaction time or movement speed based on cognitive processes, maximum effort, and functionality of the neuro-muscular system is defined as rapidity or speed [1]. Considering this, speed is a multidimensional motor ability, which manifests through three basic forms: reaction time (i.e., the ability to react to a given stimulus as fast as possible), single movement speed (i.e., the ability to reach a maximum speed of movement without resistance), and speed endurance (i.e., the ability to repeatedly produce a high speed of movement with a minimum resting period between individual repetitions) [4].

An athlete’s or police officer’s ability to quickly and accurately perceive relevant information facilitates decision making and allows more time for preparation and organization of motor behavior [5]. Previous studies of neurons reveal that there is both anatomical and physiological evidence for hierarchical information processing in the visual system. Neurons at lower levels of the visual pathway are highly specialized for simple attributes. Neurons in higher anatomical visual areas, however, generalize over these stimulus variables and are sensitive to increasingly more complex aspects of the stimuli [6]. Whether the subject is asked to make a simple or a complex response [7] differences in reaction time between these types of stimuli will persist.

Successful performance of certain critical tasks in police officers depends on the speed of reaction and the way how fast they perform the movement to respond to visual stimuli (i.e., movement response time) [3]. Some of the tasks, such as controlling a suspect, using physical force and means of coercion, handcuffing, and speed of firearms use, are also part of the high-risk tasks that officers perform [8]. Their effectiveness depends on the movement response time to previously known and unknown visual signals (i.e., the time needed from the visual signal or information to act until the end of the movement) [9]. All these tasks can be less or more complex, which could be further reflected in the speed and efficiency of the task performance. For example, a police officer could be in a situation to confront one or more offenders in which he or she must effectively and accurately act or react to offenders’ actions. Considering that the movement planning and control may be viewed as decision making, the time to embody the decision is likely to be influenced by the complexity of offenders’ actions [10,11,12].

Because of all of this, the specific physical preparation of police officers should include those activities that provide the prerequisites for effective action in given situations [3,13,14]. Thus, ability-based training should be used in the physical preparation of police officers [15,16,17]. However, testing and training the reaction time and movement speed require certain equipment and logistical conditions that police agencies often do not have [18]. With the development of modern technology more attention is put on improving the conditions for testing the speed of reaction, single movement response time as well as on neuromuscular abilities of athletes and employees in certain sectors. Therefore, the main goal of this study was to identify the effects of task and signal complexity on the single movement response time of upper and lower limbs of police officers. It was hypothesized that the movement response time of hands and legs would increase with an increase in task complexity.

## 2. Materials and Methods

### 2.1. Study Design

For this study, the response time of hands and legs was assessed in three tasks that differed by complexity. To assess the hand response time in the simplest task, participants were instructed when the light signal turns on to turn it off as fast as possible by tapping the light sensor with the dominant hand. The same protocol was followed for the leg response time in the simplest task, but the light sensor was positioned between the legs, and participants used their legs to turn off the light signal (see Appendix A). The second task by complexity started the same as the simplest, but participants were allowed to turn off the light with free choice of their hand in a hand movement task and their leg in a leg movement task (see Appendix B). The most complex task included two light signals, blue and red. Participants were instructed to use their right hand to turn the light off if it turns blue and to use their left hand to turn it off if the light turns red (see Appendix C). The same instruction was followed for the leg movement response times (see Appendix A, Appendix B and Appendix C). Prior to commencing the test, participants performed three familiarization trials. Participants were tested one by one. All tests were completed on the same day, whereby participants performed the hand tasks first. The order of tests was from the simplest to the most complex. Since we did not use a standardized test, each test was performed six times, and the best trial was used for the analysis. This way we further minimized the possibility of the learning effect. The results were recorded in milliseconds (ms).

These tests were used because they represent simple movements relatable to those occurring during police tasks. For instance, an officer could be required to reach for their teaser, baton, or weapon. Moreover, during a physical conflict, a quick block of the opponent’s strike may need to be performed. These movements are typically performed after an officer receives visual information and decides to act or react. Based on the complexity of the situation, the time to reach the movement’s terminal position may change. Although tests that mimic certain tasks could be a better choice, their standardization requires a long process. The tasks we used have defined trajectories, do not require coordination, and are easy to use across participants with different training histories. Furthermore, the complexity levels mimic the possible scenarios of a police job. For instance, an officer could have a piece of equipment in the non-dominant hand, and he/she would need to act as quickly as possible with the dominant hand or an officer could be in a position to choose which hand or leg he/she will use to act/react to the offender’s actions. In addition, an officer could be in a position to reach for the specific piece of equipment with the left or right hand or move the left or right leg based on the specific visual information he/she receives during the task.

### 2.2. Participants

The convenience sample consisted of 32 healthy male police officers aged 33.5 ± 5.5 years (range 23 to 50). The mean body height was 174.6 ± 6.3 cm (range 164 to 188 cm) and the mean body weight was 75.1 ± 6.3 kg (range 54 to 99 kg). All participants had passed a medical examination prior to testing, and they did not have any cardiovascular or neurological condition. They also had passed a physical fitness assessment at the 70th percentile of annual fitness standards and had a normal color vision and none reported difficulty seeing the visual stimuli. The purpose of the study was explained to all participants. All participants signed written informed consent. The study was carried out in accordance with the conditions of the Declaration of Helsinki and with the approval of the Ethics Committee of the Faculty of Sport and Physical Education, University of Belgrade (number 484-2).

### 2.3. Hand and Leg Movement Speed

Hand and leg movement response times were recorded using the BlazePod system (Play Coyotta Ltd., Tel Aviv, Israel). BlazePod is a system based on a wireless lighting system consisting of eight LEDs and a central PDA controller. Light signals could be either turned on or switched off by a participant or tester depending on the task. We used an interplay of the light and the task to manipulate the task complex. The BlazePod system assessed the movement speed of participants’ hands and legs in three incrementally complex tasks. The reliability and validity of the BlazePod system were determined elsewhere [19,20].

Participants started the hand movement response time tests sitting in the chair, with both their hands on the table 40 cm apart, palms facing down, and elbows at 90°. The BlazePod was located in the middle, 20 cm far apart from each hand. When the pod turned the light on, the participants were required to turn the light off as quickly as possible by touching the pod. Turning the light on was randomized so participants could not act in an expected manner but rather had to process the information and act accordingly each time the light turned on. Leg movement response time tests participants completed from the same starting position, with their feet positioned on the floor parallelly to each other, 40 cm apart with the knee angle at about 90°. The BlazePod was positioned halfway between participants’ feet. Turning the light signal on and off was performed the same way as with hands.

### 2.4. Variables

#### 2.4.1. Hand Movement Response Times

Dominant hand—hand movement response time on the light signal, expressed in ms;Free choice—hand movement response time on the light signal with free choice of either dominant or non-dominant hand, expressed in ms;Light signal—hand movement response time on the specific light signal. With right hand if the signal turns red and with left hand if signal turns blue, expressed in ms.

#### 2.4.2. Leg Movement Response Time

Dominant leg—leg movement response time on the light signal, expressed in ms;Free choice—leg movement response time on the light signal with free choice of either dominant or non-dominant hand, expressed in ms;Light signal—leg movement response time on the specific light signal. With right leg if the signal turns red and with left leg if the signal turns blue, expressed in ms.

### 2.5. Statistical Analysis

Statistical analyses were performed in JASP version 0.16.1 (University of Amsterdam, Amsterdam, The Netherlands). Descriptive statistics for mean, standard deviation, minimum, and maximum were calculated for central tendency and data dispersion. Shapiro–Wilk test was used to test the normality of data dispersion and it showed that there was no significant (*p* = 0.062–0.756) deviation from the normal distribution in all analyzed variables of movement response time. Repeated measure ANOVA was used to investigate the effects of task complexity on the single movement response time of hands and legs. The Bonferroni post hoc analysis was used for the pairwise comparison. The significance level was set at *p* < 0.05. Cohen’s effect size (d) calculation was used to determine the magnitude of effects of task complexity on movement response time and it was defined as follows: d < 0.2 (trivial or no effect), d = 0.2–0.5 (small), d = 0.5–0.8 (moderate), d = 0.8–1.3 (large), and d >1.3 (very large) [21].

## 3. Results

From the descriptive statistics (Table 1), it could be observed that the mean movement response time of the dominant hand was shorter than the mean movement response time when participants had a free choice of hand or when they acted on a specific light signal. The same trend could also be observed in leg movement response times.

The repeated measure ANOVA determined the significant effect of task complexity on a hand (Figure 1) and a leg (Figure 2) response time. Pairwise comparison for differences in movement response time of hands showed a gradual decrease in movement response time as the task complexity increased. The dominant hand response time was significantly shorter compared to the free choice of hand and movement response time on the light signal. The movement response time with a free choice of hand was significantly shorter than the movement response time on the specific light signal.

Cohen’s effect size (Table 2) indicated moderately faster movement when performed with the dominant hand compared to the free choice of hand task. Further analysis of effect sizes revealed a very large difference between the movement response time attained with the dominant hand and that obtained on the specific light signal, while a large difference occurred between the free choice and movement response time on the specific light signal.

Like in hand movement response time, the pairwise comparison for differences in movement response time of legs showed a gradual decrease in response time as the task complexity increased (Figure 2). Dominant leg movement response time was significantly shorter than the response time on the specific light signal. Further, the movement response time with a free choice of leg was significantly faster compared to that attained on a specific light signal.

Considering the effect sizes (Table 3), a small difference occurred between the dominant and free choice of leg, while a very large difference occurred when the dominant and free choice of leg movement response time was compared to the movement response time on the specific light signal.

## 4. Discussion

The main findings of this study showed that the task complexity significantly affected movement response time of upper and lower limbs as the hand and leg response times were longer when the task complexity was higher. Therefore, the hypothesis of this research is true. Considering the partial differences in movement response times between different tasks, performing the movement with the dominant hand was, while with the leg was not significantly faster compared to the free choice of hand and leg. The effect size analysis indicated a moderate difference when the tasks were performed by hands and a small difference when performed by legs, thus clearly indicating the trend of slower performance by both upper and lower limbs. This trend continued when officers performed the most complex task that resulted in a significantly longer movement response time of very large effect size when compared to dominant hand and leg, large effect size when compared to free choice of hand, and very large effect size when compared to free choice of leg.

Our results were in accordance with Hick’s law where an increment in the number of alternative choices to a stimuli increases the response time [22]. Schmidt and Lee defined two possible paradigms where the stimulus–response properties determine the success, and these are simple reaction time and choice reaction time [23]. This explains why participants responded faster on the simple task with only one choice compared to the second and third task where the number of choices increased. Considering the severity of consequences that a slow or too fast action of a police officer may have on the suspect, officer, or surrounding public, understanding the relationship between the task complexity and movement response time and reaction time is of utmost importance. For instance, Blair [3] investigated whether officers who held their firearm ready to shoot were able to shoot fast enough if a suspect who kept their gun down decided to shoot. The authors showed that the officers were generally not able to fire before the suspect if a suspect decided to shoot. Therefore, even in a simple task, an officer could be insufficiently fast. A study conducted by Mudric [24] on karate athletes investigated the simple and choice reaction times in defense movements. The authors found that athletes were faster when instructed to defend with Nagashi uke from Mae-ashi geri and with Gedan barai from Gjaku zuki (i.e., simple reaction time) compared to when they were attacked randomly with either Mae-ashi geri or Gjaku zuki (i.e., choice reaction time).

It is well known that motor planning and motor execution are mediated by hemispheric-specific processing [25,26,27,28,29]. Several studies have reported a right-hand/left-hemisphere advantage for movement execution [30,31,32], whereas a left-hand/right-hemisphere advantage has been observed for movement preparation [33,34,35]. The hemispheres of the cerebrum are specialized for different tasks. The left hemisphere is regarded as the verbal and logical brain, and the right hemisphere is thought to govern creativity, spatial relations, face recognition, and emotions, among other things. Moreover, the right hemisphere controls the left hand, and the left hemisphere controls the right hand [36]. Conversely, when training of movement tasks is sufficiently complex (i.e., skill training) to activate the motor cortex, it induces synaptogenesis, synaptic potentiation, and reorganization of movement representations within the motor cortex, thereby inducing experience-specific patterns [37]. To that end, our results indicate that officers should train in a variety of specific situations that are sufficiently challenging by complexity to develop experience-specific patterns, which in return may improve movement efficacy (e.g., improve response times and/or reduce the error rate) in variety of similar tasks [10,11,38].

Considering this and the fact that, in our study, the movement response time included the reaction time and the time to move the hand or leg to the sensor that turns off the light, the results clearly showed that as the task required more complex processing (i.e., turning only one or both hemispheres of the brain), the time to turn off the light was longer. It is worth noting that the movement response time also depends on physiological (e.g., muscle contraction and bioenergetics) and biomechanical (e.g., starting joint angle, moment arm, and hand weight). However, in our study, the conditions were standardized so the effects of these factors were minimalized or neutralized (e.g., instruction was as fast as possible and from a starting joint angle of 90°). Therefore, the differences obtained in our study are based on the effects of task complexity on the speed of neural processing and muscle contraction rather than on biomechanical factors. This is of importance when planning training for police officers as speed training may increase the movement response time but not necessarily the speed of processing and vice versa. Specific training that facilitates sensory and movement coordination (i.e., situational training) may be more beneficial [17].

### Limitations

This study included only male officers and future studies should include both males and females. The sample could have been larger. The age range could have been narrower, while the future study could include a few subsamples of officers who fall into certain age categories. The tests included simple reaction stimuli (reaction to pod light) rather than the specific cues or specific situations in which a police officer could be in. In addition, although the tasks we used were very simple and the results were very consistent, the tasks could be tested for validity and reliability.

## 5. Conclusions

Our study showed that an increment in task complexity resulted in slower performance of a single movement with a hand or a leg, suggesting an increased processing time as the identical task was performed slower when the amount of information an officer needed to process increased. This information is of importance for police officers and agencies, considering the possible repercussions of too slow processing and embodying decisions in certain police tasks. Therefore, utilization of specific and situational tasks in training is of utmost importance, as it helps the inhibition of rooted behaviors, focuses attention in a strategic way, and organizes our thoughts in terms of complexity. This notion may not be a novelty for police academies and specialized police units. However, regular police officers, who often do not have organized any kind of strength and condition or self-defense training by the agency on a regular basis (i.e., 2–3 times per week), may be at a disadvantage if found in physical conflict situations. Future studies should investigate the effectiveness of focused training interventions in occupational settings on officers’ ability to respond quickly and effectively to complex job-specific tasks.

## Figures and Tables

**Figure 1 ijerph-19-08695-f001:**
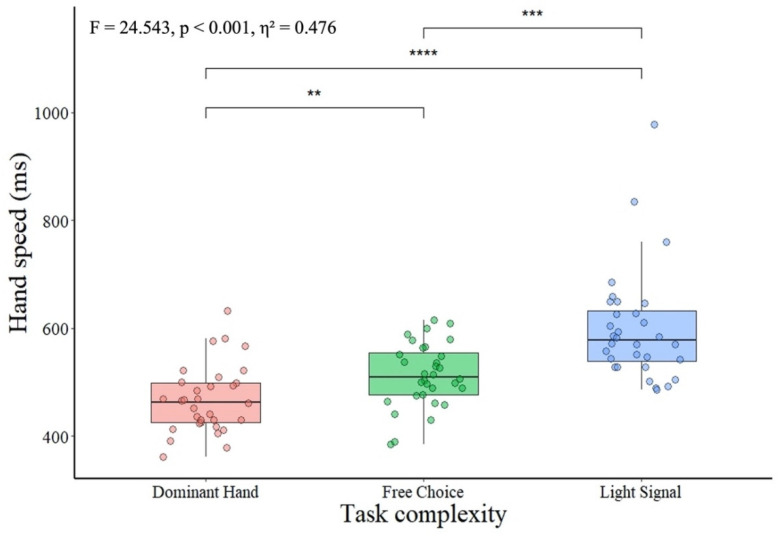
The hand movement response time in tasks of different complexity. Note: ** significant at *p* < 0.01, *** significant at *p* < 0.001, and **** significant at *p* < 0.0001. Dominant hand—response time of dominant hand on light signal; free choice—hand response time when participants were given a free choice of either dominant or non-dominant hand; light signal—hand response time on the specific light signal. With the right hand if the signal turns red and with the left limb if the signal turns blue.

**Figure 2 ijerph-19-08695-f002:**
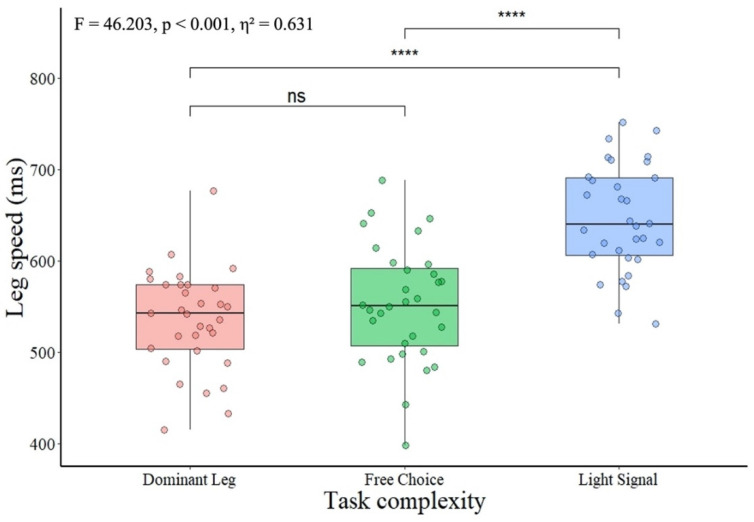
The Leg movement response time in tasks of different complexity. Note: ns—not significant and **** significant at *p* < 0.0001. Dominant hand—speed of dominant hand on light signal; dominant leg—dominant leg response time on light signal; free choice—leg response time when participants were given a free choice of either dominant or non-dominant leg; light signal—leg response time on the specific light signal. With the right leg if the signal turns red and with the left leg if the signal turns blue.

**Table 1 ijerph-19-08695-t001:** Descriptive statistics.

Variables	Mean	Standard Deviation	Minimum	Maximum
Age	33.5	5.5	23.0	50.0
Height	174.6	6.3	164	188
Weight	75.1	11.9	54	99
Hand speed				
Dominant hand (ms)	467.026	62.353	360.500	631.667
Free choice (ms)	512.740	59.022	384.833	615.000
Light signal (ms)	599.099	103.293	485.000	977.167
Leg speed				
Dominant leg (ms)	535.177	54.532	414.667	676.167
Free choice (ms)	552.630	63.924	397.500	688.167
Light signal (ms)	646.229	58.571	531.167	751.333

Note. Dominant hand—response time of dominant hand on light signal; dominant leg—dominant leg response time on light signal; free choice—limb response time when participants were given a free choice of either dominant or non-dominant hand in hand task and leg in leg task; light signal—limb response time on the specific light signal. With the right limb if the signal turns red and with the left limb if the signal turns blue.

**Table 2 ijerph-19-08695-t002:** Effect of decision process on single movement response time of the dominant arm.

Pairwise Comparison		Mean Diff.	95% Confidence Int.		Std. E.	T	*d*
			Lower	Upper			
Dominant hand	Free choice	−42.571	−85.641	−12.767	12.476	−2.673	−0.559
	Light signal	−127.643	−170.713	−84.573	19.998	−8.015	−1.692
Free choice	Light signal	−85.071	−128.141	−42.002	22.239	−5.342	−1.133

Note: Mean Diff.—mean difference, Confidence Int.—confidence interval, Std. E.—standard error, *d*—Cohen’s effect size. Dominant hand—response time of dominant hand on light signal; free choice—hand response time when participants were given a free choice of either dominant or non-dominant hand; light signal—hand response time on the specific light signal. With the right hand if the signal turns red and the with left hand if the signal turns blue.

**Table 3 ijerph-19-08695-t003:** Effect of task complexity on leg movement response time.

Pairwise Comparison		Mean Diff.	95% Confidence Int.		Std. E.	T	*d*
			Lower	Upper			
Dominant leg	Free choice	−11.071	−38.972	16.829	10.316	−1.073	−0.203
	Light signal	−106.536	−134.436	−78.635	10.316	−10.327	−1.952
Free choice	Light signal	−95.464	−123.365	−67.564	10.316	−9.254	−1.749

Note: Dominant leg—response time of dominant leg on light signal; free choice—leg response time when participants were given a free choice of either dominant or non-dominant leg; light signal—leg response time on the specific light signal. With the right leg if the signal turns red and with the left leg if the signal turns blue.

## Data Availability

Available upon request. Contact: filip.kukic@gmail.com.

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
