# Peer review of "Effects of the Task Complexity on the Single Movement Response Time of Upper and Lower Limbs in Police Officers"

_ijerph, 2022, doi:10.3390/ijerph19148695_

Round 1

Reviewer 1 Report

Dear Authors,

Submitted for review the manuscript was not written according to basic requirements for scientific papers. The manuscript contains many editorial faults (errors).

Detailed comments:

1.    Title: inconsistent with the tasks (trials) carried out (speed). The Authors carried out the research of reaction time (psychomotor performance, reaction time, complex reaction time), and not speed. Please note that there is a difference between reaction time and speed.

2.    Key words: words ʺBlaze Podʺ and ʺtactical athletesʺ seem inappropriate.

Introduction

3.    Compare Line 73 ʺthe main goal of…” and Line „The purpose of the study…”. What is the difference between „the goal” and ʺthe purpose”? Study require unequivocal terms (definitions).

4.    Line 50 – [7] , [8]. Why was not the entry [7, 8] applied? Similar line 40.

5.    The introduction must be supplemented with articles refer to similar topics (policemen, soldiers). I suggest soldiers, as long as there is no research relating to policemen and some tasks are similar.

6.    The aim of research is inconsistent with the tasks (trials) carried out (speed). The Authors carried out the research of reaction time and not speed. What is a difference between your research and research others researchers refer to reaction time? Physical fitness tests in which we can measure speed can be found e.g. in International Physical Fitness Test and EUROFIT.

Materials and Methods

7.    2.1. Study design: There is no information on who is the author of the tasks (test; trials?) you used. There is no justification why the authors chose these tasks / trials.

8.       Line 97 (range 54 to 9 kg) – wrong entry.

9.       2.4. Variables. Was it noted which limb was chosen by the respondents (dominant or non-dominant) during task? Maybe in the task 2 respondents chose the dominant limb (just as task 1).

Results

10.     Line 165 ʺThe Cohen`s effect size (Table 1) indicated…”. Table 1 does not show findings of Cohen`s effect size.

11.     Table 2 and Table 3 – please explain why three variables were entered (ʺFree choice”, ʺSpecific signal”, ʺSpecific signal”)? These entries (descriptions) are incomprehensible to the reader. Maybe it is worth adding information under the table.

12.     Table 1, the result ʺ0.498” – please check that it is correct. This result is clearly different from the other data in this table.

Discussion

13.     Line 186 - ʺTherefore, the main hypothesis of this…”. Why was ʺthe main hypothesis” written? There is only one hypothesis in the manuscript (see: Introduction).

14.     The discussion requires supplementing and comparing the research results with other current research results.

References

15.     The selection of literature is insufficient. There are very few articles about policemen. This manuscript could use the results of studies conducted among soldiers during military training or SERE. In PubMed and Google browsers, it is enough to enter the key words: simple reaction time, Multiple Choice Reaction Time (MCRT), divided attention, soldiers, SERE, survival, military training.

16.     The references were developed (elaborate) not in accordance with the requirements of the editorial office (Journal Articles: 1. Author 1, A.B.; Author 2, C.D. Title of the article. Abbreviated Journal Name Year, Volume, page range).

17.     The cited articles are obsolete, e.g. No 2 of 1975, No 4 of 1993, No 5 of 1991, No 6 of 1985, No 7 of 1991, No 8 of 1987, No 10 from 1954 etc.

About 45% of the cited articles are from before 2000, i.e. 22 years ago. I am asking the authors to justify such a selection of literature.

Author Response

Dear Authors,

Submitted for review the manuscript was not written according to basic requirements for scientific papers. The manuscript contains many editorial faults (errors).

Detailed comments:

  1. Title: inconsistent with the tasks (trials) carried out (speed). The Authors carried out the research of reaction time (psychomotor performance, reaction time, complex reaction time), and not speed. Please note that there is a difference between reaction time and speed.

Reply: We understand the reviewer’s concern. However, respectfully, the tasks we carried out included movement. We measured time from the signal until the end of the movement. The reaction time in our task would be only from the moment the signal turns on until the participant makes the first move but does not complete the movement. Although the movement was short and simple, the work and strength could be calculated. In reaction time, there is no mechanical work or strength. Also, considering the journal title and that the results of this study could be of interest to a non-sport-specific audience, the simple movement speed is practical terminology understandable to a readership outside of sport and exercise science.

  1. Key words: words ʺBlaze Podʺ and ʺtactical athletesʺ seem inappropriate.

Reply: We removed “Blaze Pod” and “tactical athletes”.

Introduction

  1. Compare Line 73 ʺthe main goal of…” and Line „The purpose of the study…”. What is the difference between „the goal” and ʺthe purpose”? Study require unequivocal terms (definitions).

Reply: With all due respect, we do not think the goal and the purpose are the same. A goal is a point one wishes to achieve. On the other hand, purpose can be called the reason one aims at to achieve a goal. We used “goal” in the last paragraph of the introduction to set the final destination, while we used the “purpose” in the participants section because it encompasses the goal and the explanation of its meaning so participants understand why are they asked to participate.

  1. Line 50 – [7] , [8]. Why was not the entry [7, 8] applied? Similar line 40.

Reply: Thank you for noticing this. We fixed it.

  1. The introduction must be supplemented with articles refer to similar topics (policemen, soldiers). I suggest soldiers, as long as there is no research relating to policemen and some tasks are similar.

Reply: We certainly agree with the reviewer that referring to similar studies on similar populations is important. However, we could not find studies on these populations that relate well to our study design. While this may rise some concerns, at the same time it provides an important space for investigation. To that end, we added a few more references that further strengthen the theoretical and scientific foundations for our study.

  1. The aim of research is inconsistent with the tasks (trials) carried out (speed). The Authors carried out the research of reaction time and not speed. What is a difference between your research and research others researchers refer to reaction time? Physical fitness tests in which we can measure speed can be found e.g. in International Physical Fitness Test and EUROFIT.

Reply: We understand the reviewers’ concern. However, the tasks performed in this study included movement. We are aware that these movements were simple and short and the results could be caused by reaction time, but reaction time was not measured so we felt that it would rather be speculation. “The most reliable method is to use an extrapolation of the change in the average force that the finger exerts on the surface to estimate the reaction time.” Brenner E, Smeets JBJ. How Can You Best Measure Reaction Times? J Mot Behav. 2019;51(5):486-495. doi: 10.1080/00222895.2018.1518311. Epub 2018 Oct 25. PMID: 30358504. Also, check this study on martial arts athletes https://www.ncbi.nlm.nih.gov/pmc/articles/PMC3863923/ that investigated reaction time and movement speed (i.e., 25 cm long movement). The authors found no difference in reaction times (both simple and choice reaction times), while martial artists had faster simple movement speed compared to controls.

There is a scarcity of research with police regarding the single movement speed, especially on the effects of the task complexity on single movement speed. Considering the study by Blair et al. 2011 (https://journals.sagepub.com/doi/abs/10.1177/1098611111423737) who found that police officers tend to react slower than the armoured suspect, our study is of importance as it shows that even in simple task small additions to complexity affect the time needed to the movement to reach its final destination (i.e., movement speed).   

Speed tests such as 50m sprint (International Physical Fitness Test) and Plate Tapping (EUROFIT) do not provide the answer to our study question. It would be best to mimic the police task, which we are planning on doing. However, obtaining approval for the study protocol, equipment, and facilities is a long process.

Materials and Methods

  1. 1. Study design: There is no information on who is the author of the tasks (test; trials?) you used. There is no justification why the authors chose these tasks / trials.

Reply: We are the authors of the tasks. We wanted to use a simple task that will show the effects of increment in complexity on task performance. We expanded the first paragraph of the methods section with explanations of why these tasks were used.

This is added: These tests were used because they represent simple movements relatable to those occurring during police tasks. For instance, an officer could be required to reach for the teaser, baton or weapon. Also, during a physical conflict, a quick block of the opponent’s strike may need to be performed based. These movements are typically performed after an officer receives visual information and decides to act or react. Based on the complexity of the situation, the time to reach the movement’s terminal position may change. Although tests that mimic certain tasks could be a better choice, their standardization requires a longer process to be standardized. The tasks we used have a defined trajectory, do not require coordination, and are easy to use across participants with different training histories. Furthermore, the complexity levels mimic the possible scenarios of a police job. For instance, an officer could have a piece of equipment in the non-dominant hand and he/she would just need to act as quickly as possible with the dominant hand or an officer could be in a position to choose which hand or leg he/she will use to act/react to the offender’s actions. In addition, an officer could be in a position to reach for the specific piece of equipment with the left or right hand or move the left or right leg based on the specific visual information he/she receives during the task.

  1. Line 97 (range 54 to 9 kg) – wrong entry.

Reply: Thank you for noticing this. Changed in text 54 to 99kg

  1. Was it noted which limb was chosen by the respondents (dominant or non-dominant) during task? Maybe in the task 2 respondents chose the dominant limb (just as task 1).

Reply: Thank you for raising this question. We did not mark which hand/leg they used in the second task. However, in the context of our study goal, this is of less importance. The point we were trying to make is that the time to perform the task, even a very simple one, will prolong when participants are given choice. Whether they will perform with a dominant or non-dominant hand was not in our focus. This, however, would be an interesting topic to investigate in another study.

Results

  1. Line 165 ʺThe Cohen`s effect size (Table 1) indicated…”. Table 1 does not show findings of Cohen`s effect size.

Reply: Thank you for pointing this out. We fixed it.

  1. Table 2 and Table 3 – please explain why three variables were entered(ʺFree choice”, ʺSpecific signal”, ʺSpecific signal”)? These entries (descriptions) are incomprehensible to the reader. Maybe it is worth adding information under the table.

Reply: We appreciate your concern. We added the legend below the table to explain the terminology. We inserted these explanations in each Table and Figure.

This is added: Note. Dominant hand – speed of dominant hand on light signal; Dominant leg – dominant leg speed on light signal; Free choice – limb speed when participants were given a free choice of either dominant or non-dominant hand in hand task and leg in leg task; Light signal – limb speed on the specific light signal. With the right limb, if the signal turns red and with the left limb if the signal turns blue.

  1. Table 1, the result ʺ0.498” – please check that it is correct. This result is clearly different from the other data in this table.

Reply: Thank you very much for noticing this. We revised the statistics and fixed the table accordingly. 

Discussion

  1. Line 186 - ʺTherefore, the main hypothesis of this…”. Why was ʺthe main hypothesis” written? There is only one hypothesis in the manuscript (see: Introduction).
  2. Reply: We removed “main” from this sentence.
  3. The discussion requires supplementing and comparing the research results with other current research results.

Reply: In general, we agree with this suggestion. However, other than the studies we already cited in the discussion, we did not find similar studies in the tactical population. We did, however, strengthened our findings with additional references that support our results.

References

  1. The selection of literature is insufficient. There are very few articles about policemen. This manuscript could use the results of studies conducted among soldiers during military training or SERE. In PubMed and Google browsers, it is enough to enter the key words: simple reaction time, Multiple Choice Reaction Time (MCRT), divided attention, soldiers, SERE, survival, military training.

Reply: We added more references about the policemen

  1. https://pubmed.ncbi.nlm.nih.gov/30507731/
  2. https://www.ncbi.nlm.nih.gov/pmc/articles/PMC6102190/
  3. https://www.frontiersin.org/articles/10.3389/fpsyg.2019.01797/full
  4. https://journals.lww.com/nsca-scj/Abstract/2021/12000/Occupational_Challenges_to_the_Development_and.13.aspx
  5. https://pubmed.ncbi.nlm.nih.gov/31045682/
  6. https://www.mdpi.com/2313-576X/5/3/54/htm

  1. The references were developed (elaborate) not in accordance with the requirements of the editorial office (Journal Articles: 1. Author 1, A.B.; Author 2, C.D. Title of the article. Abbreviated Journal NameYearVolume, page range).
  2. Reply: Thank you for noticing this. It is fixed now.

  1. The cited articles are obsolete, e.g. No 2 of 1975, No 4 of 1993, No 5 of 1991, No 6 of 1985, No 7 of 1991, No 8 of 1987, No 10 from 1954 etc.

Reply: We have replaced references of an older date with new ones

  1. About 45% of the cited articles are from before 2000, i.e. 22 years ago. I am asking the authors to justify such a selection of literature.

Reply: References older than 20 years have been replaced

Reviewer 2 Report

1. author Iva Prćić has affiliation number 4, but affiliation 4 is not shown in the paper

2. Lines 31-33 - that sentence requires a quote

3. Line 51: 60; 199 - when quoting then stating the number of pages is unnecessary (pp 73-28; p44....)

4. Line 84 - Insert Appendix A instead of Figure A1

5. Lines 86-87 - Insert Appendix B instead of Figure A2

6. Line 89 - Insert Appendix C instead of Figure A3

7. Line 94 - Insert Appendix A, B and C

8. Lines 90-93 - Quote from when the tests were taken? Are the tests standardized? Why are they performed 6 times? Is that according to the protocol? All this needs to be explained.

9. Line 96 - There is a large age range between respondents (23-50)! This is one of the reasons why differences in the speed of performing different tasks can occur. Could there have been a smaller age difference among the respondents?

10. Line 96 - Here the range is from 162 to 188 cm, but in table 1 is 164 to 188?

11. Line 97 - Insert 99 kg instead of 9 kg

12. Lines 128-129; 134-135 - Appendix C shows otherwise?

13. Lines 154-156 - these results need to be shown in the table.

14. Table 3 - There should not be the same legend under Table 3 as under Table 2, it is unnecessary

15. Lines 187-189 -  Figure 1 shows that performing the movement with the Dominant hand  was significantly faster compared to Free choice of  hand?

16. Lines 242-244 - Add more limitations of the study, the age difference between respondents, is a small sample, maybe use some situational tests specific to this type of work...

Author Response

  1. author Iva Prćić has affiliation number 4, but affiliation 4 is not shown in the paper

Reply: Thank you for pointing this out. We added affiliation: Institute of Medical Research, Belgrade University,11129 Belgrade, Serbia; [email protected] (I.P);

  1. Lines 31-33 - that sentence requires a quote

Reply: We added the reference.

  1. Line 51: 60; 199 - when quoting then stating the number of pages is unnecessary (pp 73-28; p44....)

Reply: This is fixed throughout the text.

  1. Line 84 - Insert Appendix A instead of Figure A1

Reply: We fixed it accordingly.

  1. Lines 86-87 - Insert Appendix B instead of Figure A2

Reply: We fixed it accordingly.

  1. Line 89 - Insert Appendix C instead of Figure A3

Reply: We fixed it accordingly.

  1. Line 94 - Insert Appendix A, B and C

Reply: Inserted.

  1. Lines 90-93 - Quote from when the tests were taken? Are the tests standardized? Why are they performed 6 times? Is that according to the protocol? All this needs to be explained.

Reply: We are the authors of the tasks. We wanted to use a simple task that will show the effects of increment in complexity on task performance. We expanded the first paragraph of the methods section with explanations of why these tasks were used.

This is added: These tests were used because they represent simple movements relatable to those occurring during police tasks. For instance, an officer could be required to reach for the teaser, baton or weapon. Also, during a physical conflict, a quick block of the opponent’s strike may need to be performed. These movements are typically performed after an officer receives visual information and decides to act or react. Based on the complexity of the situation, the time to reach the movement’s terminal position may change. Although tests that mimic certain tasks could be a better choice, their standardization requires a long process. The tasks we used have defined trajectories, do not require coordination, and are easy to use across participants with different training histories. Furthermore, the complexity levels mimic the possible scenarios of a police job. For instance, an officer could have a piece of equipment in the non-dominant hand and he/she would need to act as quickly as possible with the dominant hand or an officer could be in a position to choose which hand or leg he/she will use to act/react to the offender’s actions. In addition, an officer could be in a position to reach for the specific piece of equipment with the left or right hand or move the left or right leg based on the specific visual information he/she receives during the task.

  1. Line 96 - There is a large age range between respondents (23-50)! This is one of the reasons why differences in the speed of performing different tasks can occur. Could there have been a smaller age difference among the respondents?

Reply: We appreciate the reviewer’s concern and agree that in ideal conditions, a more narrow age range would be more convenient. We added this in Limitations. However, in occupational settings such as police recruiting participants is difficult. Especially when the study wasn’t ordered from higher instances in the chain of command but organized from the lower instances such as our study. Considering all this, we still had sufficiently consistent results that clearly show the effects of task complexity on movement speed.   

This is added: The age range could have been narrower, while the future study could include a few subsamples of officers who fall into certain age categories.

  1. Line 96 - Here the range is from 162 to 188 cm, but in table 1 is 164 to 188?

Reply: Thank you for noticing this. It is fixed now.

  1. Line 97 - Insert 99 kg instead of 9 kg

Reply: Thank you for noticing this. It is fixed now.

  1. Lines 128-129; 134-135 - Appendix C shows otherwise?

Reply: No. Appendix C represents this.

  1. Lines 154-156 - these results need to be shown in the table.

Reply: We appreciate the reviewer’s suggestion. While this information does not fit the table, we added it to Figures 1 and 2.

  1. Table 3 - There should not be the same legend under Table 3 as under Table 2, it is unnecessary

Reply: We removed the repeating legend.

  1. Lines 187-189 -  Figure 1 shows that performing the movement with the Dominant hand  was significantly faster compared to Free choice of  hand?

Reply: Yes

  1. Lines 242-244 - Add more limitations of the study, the age difference between respondents, is a small sample, maybe use some situational tests specific to this type of work...

Reply: We added the age range and the sample size in Limitations. Situational tests were already mentioned.

Reviewer 3 Report

Review

Effects of the task complexity on the single movement speed of upper and lower limbs in police officers

Abstract

Remove “may” or “occasionally” from line 12

Line 23 – what is meant by “initial”?

Line 24 – you don’t need 3 decimal points after F values just because you have 3 for p values.

I think the last sentence could say something else besides restating the results. Something about what is might mean to the police?

Introduction

Line 33 add an s to condition

The sentence from line 33 to 35 is confusing. Can it be re-written?

Line 50 what is researches?

The references need fixed use [] for all # and don’t put page numbers in.

I suggest have the introduction examined for correct English.

Materials and Methods

A complete review is need for correct English. Line 79 – different should be “that differed…” Line 81 this sentence as written makes little sense.

Sample – small, convenience sample. Was a sample size determination made prior to the study? The effect sizes presented help.

Discussion

The third paragraph, while informative, seems to offer too much information. This could be shortened and then add in how the results of the current study fit.

Author Response

Effects of the task complexity on the single movement speed of upper and lower limbs in police officers

Abstract

Remove “may” or “occasionally” from line 12

Reply: Thank you for this suggestion. We removed “may”.

Line 23 – what is meant by “initial”?

Reply: We removed “of the initial data”.

Line 24 – you don’t need 3 decimal points after F values just because you have 3 for p values.

Reply: We shortened it to 1 decimal for F.

I think the last sentence could say something else besides restating the results. Something about what is might mean to the police?

Reply: We changed the last sentence of the Abstract with: “The training of police officers should utilize specific and situational tasks to improve movement speed by improving the redundancy in decision-making processes during the work specific tasks of different complexity.”

Introduction

Line 33 add an s to condition

Reply: This sentence is revised and does not include “conditions” anymore.

The sentence from line 33 to 35 is confusing. Can it be re-written?

Reply: Thank you for pointing this non-understandable sentence out. It has been addressed for better clarity.

Line 50 what is researches?

Reply: Replaced with “studies”

The references need fixed use [] for all # and don’t put page numbers in.

Reply: References are fixed throughout the paper.

I suggest have the introduction examined for correct English.

 Reply: The manuscript has now been reviewed.

Materials and Methods

A complete review is need for correct English. Line 79 – different should be “that differed…” Line 81 this sentence as written makes little sense.

 Reply: Thank you for the suggestion. It is fixed accordingly.

Sample – small, convenience sample. Was a sample size determination made prior to the study? The effect sizes presented help.

 Reply: The sample was not predefined. It was a convenience sample because we conducted research using one of our training groups. Also, from this training group, only those who voluntarily participated were tested. We added “convenience” in methods and a small sample size in Limitations.

Discussion

The third paragraph, while informative, seems to offer too much information. This could be shortened and then add in how the results of the current study fit.

Reply: Thank you for this suggestion. We addressed this paragraph accordingly. The first sentence is omitted and we added two sentences at the end of this paragraph.

This is added: Conversely, when training of movement tasks is sufficiently complex (i.e., skill training) to activate the motor cortex, it induces synaptogenesis, synaptic potentiation, and reorganization of movement representations within the motor cortex, thereby inducing experience-specific patterns [37]. To that end, our results indicate that officers should train in a variety of specific situations that are sufficiently challenging by complexity to develop experience-specific patterns, which in return may improve movement efficacy (e.g., increase speed and/ or reduce the error rate) in variety of similar tasks [10,11,28].

Reviewer 4 Report

It would be interesting to carry out the evaluation in a real context.

Author Response

It would be interesting to carry out the evaluation in a real context.

Reply: We totally agree with the Reviewer. This is in planning and currently, we are waiting the approval of the officers in charge of the facilities and equipment needed for the experiment.

Round 2

Reviewer 1 Report

Dear Authors,

Thank you for your response and the manuscript`s correccion. Unfortunately I cannot agree with you in basic issues.

1  1. I cannot agree with the authors that movement speed of upper and lower limbs was measured. In my opinion the authors measured reaction time (see: section 2.1.). Please let me know what is the difference between your „test” and e.g. reaction time test, (Multi) Choice Reaction Time, response time with a choice, divided attention test.

       Note what units of measurement you have used.

Speed is measured as the ratio of distance to the time in which the distance was  achieved.

Speed has the dimensions of distance divided by time. The SI unit of speed is the metre per second (m/s), but the most common unit of speed in everyday usage is the kilometre per hour (km/h). Please see what units of measurement you have in your test results. This is a millisecond (ms) (Tab. 1, Fig. 1, Fig. 2).

Reaction time may be defined simply as the time between a stimulus and a response. Serial reaction time is a combination of recognition and choice reaction.

22. The authors wrote that used their own test („This is added: These tests were used because they represent simple movements relatable to those occurring during police tasks). Each "test" must undergo an appropriate validation procedure and meet the appropriate conditions (standardization, reliability, validity, normalization). Please write where the validation procedure for this test is described.

33. How did the authors interpret the test results when the subject made a mistake (e.g. when they used a different arm / leg than was required for the task)?

4 4. The Abstract was developed incorrect (see: https://www.mdpi.com/journal/ijerph/instructions#preparation).

Author Response

Dear Authors,

Thank you for your response and the manuscript`s correccion. Unfortunately I cannot agree with you in basic issues.

1  1. I cannot agree with the authors that movement speed of upper and lower limbs was measured. In my opinion the authors measured reaction time (see: section 2.1.). Please let me know what is the difference between your „test” and e.g. reaction time test, (Multi) Choice Reaction Time, response time with a choice, divided attention test.

       Note what units of measurement you have used.

Speed is measured as the ratio of distance to the time in which the distance was achieved.

Speed has the dimensions of distance divided by time. The SI unit of speed is the metre per second (m/s), but the most common unit of speed in everyday usage is the kilometre per hour (km/h). Please see what units of measurement you have in your test results. This is a millisecond (ms) (Tab. 1, Fig. 1, Fig. 2).

Reaction time may be defined simply as the time between a stimulus and a response. Serial reaction time is a combination of recognition and choice reaction.

Reply: Yes, the reaction time may (or may not) be defined simply as the time between the stimulus and a response. To that end, we are happy to change the terminology and call it simple movement response time. Otherwise, if we put the “=” sign between the task we used (neural-muscular components of movement) and the reaction time (neural component of movement), we would need to conduct a study that would prove our test is a valid representation of reaction time. Simple movement speed or simple movement response time represent precisely what the task was – maximally fast very well-defined simple movement.

  1. 2The authors wrote that used their own test („This is added:These tests were used because they represent simple movements relatable to those occurring during police tasks). Each "test" must undergo an appropriate validation procedure and meet the appropriate conditions (standardization, reliability, validity, normalization). Please write where the validation procedure for this test is described.

Reply: We agree that validation is important. However, this was a very simple task. It does not require coordination and it does not require a skill. It can be performed by any healthy person. We had our participants familiarize and they performed the test 6 times with sufficient rest. The consistency of results is also promising. However, we added the statement regarding the validation in Limitations.

  1. 3How did the authors interpret the test results when the subject made a mistake (e.g. when they used a different arm / leg than was required for the task)?

Reply: Mistakes were not interpreted nor analyzed in this study. In the third task, if participants used the wrong hand for a given signal, the repetition was not counted. They repeated the test until they had 6 successful trials. The fasted trial was used for the analysis. The effect of task complexity on the likelihood of making mistakes was not the focus of our study. This is an important issue that could be addressed in a separate study.

4 4. The Abstract was developed incorrect (see: https://www.mdpi.com/journal/ijerph/instructions#preparation).

Reply: We do not see an issue in the development of the abstract. We do have more than 200 words but the extension is made per the request from another reviewer.